# Apoptotic Body-Rich Media from Tenocytes Enhance Proliferation and Migration of Tenocytes and Bone Marrow Stromal Cells

**DOI:** 10.3390/ijms231911475

**Published:** 2022-09-29

**Authors:** Chenhui Dong, Anne Gingery, Peter C. Amadio, Kai-Nan An, Steven L. Moran, Chunfeng Zhao

**Affiliations:** 1Biomechanics & Tendon and Soft Tissue Biology Laboratories, Division of Orthopedic Research, Mayo Clinic, Rochester, MN 55905, USA; 2Department of Sports medicine, The 940th Hospital of Joint Logistics Support Force of PLA, Lanzhou 730050, China

**Keywords:** tenocytes, bone marrow stromal cells, proliferation, migration, apoptotic body

## Abstract

The intrinsic healing following tendon injury is ideal, in which tendon progenitor cells proliferate and migrate to the injury site to directly bridge or regenerate tendon tissue. However, the mechanism determining why and how those cells are attracted to the injury site for tendon healing is not understood. Since the tenocytes near the injury site go through apoptosis or necrosis following injury, we hypothesized that secretions from injured tenocytes might have biological effects on cell proliferation and migration to enhance tendon healing. Tenocyte apoptosis was induced by 24 h cell starvation. Apoptotic body-rich media (T-ABRM) and apoptotic body-depleted media (T-ABDM) were collected from culture media after centrifuging. Tenocytes and bone marrow-derived stem cells (BMDSCs) were isolated and cultured with the following four media: (1) T-ABRM, (2) T-ABDM, (3) GDF-5, or (4) basal medium with 2% fetal calf serum (FCS). The cell activities and functions were evaluated. Both T-ABRM and T-ABDM treatments significantly stimulated the cell proliferation, migration, and extracellular matrix synthesis for both tenocytes and BMDSCs compared to the control groups (GDF-5 and basal medium). However, cell proliferation, migration, and extracellular matrix production of T-ABRM-treated cells were significantly higher than the T-ABDM, which indicates the apoptotic bodies are critical for cell activities. Our study revealed the possible mechanism of the intrinsic healing of the tendon in which apoptotic bodies, in the process of apoptosis, following tendon injury promote tenocyte and stromal cell proliferation, migration, and production. Future studies should analyze the components of the apoptotic bodies that play this role, and, thus, the targeting of therapeutics can be developed.

## 1. Introduction

Musculoskeletal injures are the leading cause of health care visits in the United States with a huge economic burden [1,2]. Tendon injuries in the hand are common and have a major impact on work and function. Despite advances in surgical repair techniques, healing of finger flexor tendons remains problematic because of poor vascularization and hypocellularity [3,4]. Flexor tendon healing often results in adhesion formation, an extrinsic healing mechanism that restricts tendon motion and reduces hand function. In contrast, if a tendon heals intrinsically, e.g., through tenocyte proliferation and migration, there will be less scar and adhesion formation and correspondingly better function [5,6,7,8]. One piece of evidence for the intrinsic healing of tendons is that the tenocytes or progenitor cells proliferate and migrate to the injury site to promote tendon healing by producing Col-1 and Col-3 [9,10,11]. However, the mechanism of these cellular activities is unclear. It is known that tenocytes at the injury site undergo apoptosis after tendon injury [12]. Apoptotic body formation occurs when cells undergo apoptosis [13,14]. This packaging of apoptotic cells is primarily thought to mediate immunologic and phagocytotic clearance. However, whether the secretions from apoptotic cells would stimulate the cell migration and proliferation of intrinsic tenocytes or progenitor cells from either the blood or bone marrow are unknown.

In this study, we investigated the effects of apoptotic bodies released by apoptotic tenocytes on freshly cultured canine flexor tendon tenocytes and bone marrow-derived stem cells (BMDSCs) regarding cell proliferation and migration. We hypothesized that tenocyte apoptotic body-rich media (T-ABRM) would promote tenocyte and BMDSC proliferation and migration when compared to tenocyte apoptotic body-depleted media (T-ABDM) and typical culture media as well. 

## 2. Results

### 2.1. Characterization of a Cell Culture Conditioning Medium

Conditioned media from apoptotic tenocytes contained 24.1% vesicles larger than 3 µm (gate R1), which were mainly suspended cells and dead cell debris, 70.3% vesicles between 1 and 3 µm (gate R2), which were mainly apoptotic bodies, and 3.6% microvesicles smaller than 1 µm (gate R3) (Figure 1A second panel). After centrifugation to obtain T-ABRM, the apoptotic body concentration increased to 85.9% (Figure 1A third panel). In the T-ABDM obtained by the centrifugation (16,000 g, 20 min) of T-ABRM, the apoptotic body concentration decreased to 35.7%. However, the vesicles smaller than 1 µm increased to 53% (Figure 1A fourth panel).

In the control group in which the cells were cultured under normal culture conditions (10% serum with minimum essential medium [MEM]), only 4.28% ± 1.2% of cells were annexin-V (AV) positive and propidium iodide (PI) negative, indicating few apoptotic cells, and 4.25% ± 0.7% were positive for both AV and PI, indicating few necrotic cells. However, after 24 h in serum deprivation (T-ABRM), 48.3% ± 2.1% of cells were AV positive and PI negative, indicating a pronounced induction of apoptosis in tenocytes, and almost all the remaining cells, 52% ± 4%, were positive for both AV and PI, indicating necrosis. In the T-ABDM, 21% were AV positive and PI negative and only 1.78% were AV and PI positive, which indicated that the majority of the apoptotic and necrotic cells were removed by high-speed centrifugation (Figure 1A,B).

Figure 1C illustrates that normal tenocytes were positive for DAPI (green) and DiI (red), with the cytoplasm staining red and the nuclei staining green on fluorescence confocal microscopy (left). After 24 h of culture without serum, significant cytoplasm shrinkage with dark and unclearly stained nuclei was observed (middle). In addition, many divided nuclei with cell membrane-covered vesicles (apoptotic bodies) were detected (right).

### 2.2. T-ABRM and T-ABDM-Mediated Cell Proliferation

The cell proliferation results showed significant differences in both tenocytes and BMDSCs incubated with T-ABRM compared to those incubated with basal media or T-ABDM alone at 24, 48, and 72 h of cultivation (*p* < 0.05) (Figure 2A,B). Tenocyte and BMDSC proliferation increased in a T-ABRM dosage-dependent manner. Interestingly, T-ABRM media in a 1:1 ratio (T-ABRM: basic-media) had the highest cell proliferation after 48 h. Cell proliferation of the tenocytes in the T-ABRM media was significantly increased as compared to BMDSCs, indicating that tenocytes increased their responsiveness to apoptotic bodies compared to BMDSCs (Figure 2C).

### 2.3. Cell Migration under T-ABRM and T-ABDM Stimulation

Tenocyte and BMDSC migration was significantly increased in both the T-ABDM and T-ABRM groups compared to the GDF-5 and control media groups (*n* = 4, *p* < 0.05) (Figure 3A,B). The migration rate of tenocytes with GDF-5 stimulation was significantly higher than that of the control group (Figure 3A), but there was no significant difference between GDF-5 and control in the BMDSC group (Figure 3B). Figure 3C–J is a representative image of the migration assays for both tenocytes (C–F) and BMDSCs (G–J) after 24 h.

### 2.4. Gene Expression Assay

Gene expression measured by quantitative reverse transcription–polymerase chain reaction (RT-PCR) revealed that in the T-ABRM-treated tenocyte group, collagen 1a (Col-1a) mRNA was significantly upregulated with culture time compared to the control group at 24 h (1.2 ± 0.3-fold, *n* = 3, *p* < 0.05), 48 h (1.9 ± 0.2-fold, *n* = 3, *p* < 0.05), and 72 h (3.3 ± 0.3-fold, *n* = 3, *p* < 0.05). Collagen 3a (Col-3a) mRNA was also significantly upregulated at 24 h (1.4 ± 0.1-fold, *n* = 3, *p* < 0.05), 48 h (2.1 ± 0.1-fold, *n* = 3, *p* < 0.05), and 72 h (2.6 ± 0.1-fold, *n* = 3, *p* < 0.01). Transforming growth factor β (TGF-β) was upregulated at 24 h (3.34 ± 0.1-fold, *n* = 3, *p* < 0.05), 48 h (2.8 ± 0.3-fold, *n* = 3, *p* < 0.05), and 72 h (1.8 ± 0.1-fold, *n* = 3, *p* < 0.05) compared to the control group (Figure 4).

In the T-ABRM-treated BMDSC group, Col-1a mRNA was upregulated compared to the control group at 24 h (3.3 ± 0.1-fold, *n* = 3, *p* < 0.05), 48 h (2.3 ± 0.1-fold, *n* = 3, *p* < 0.05), and 72 h (1.2 ± 0.1-fold, *n* = 3, *p* < 0.05). Col-3a mRNA was also significantly upregulated at 24 h (3.7 ± 0.1-fold, *n* = 3, *p* < 0.05), 48 h (4.3 ± 0.2-fold, *n* = 3, *p* < 0.05), and 72 h (2.6 ± 0.1-fold, *n* = 3, *p* < 0.01). TGF-β was upregulated at 24 h (2.9 ± 0.1-fold, *n* = 3, *p* < 0.05), 48 h (2.0 ± 0.2-fold, *n* = 3, *p* < 0.05), and 72 h (3.2 ± 0.1-fold, *n* = 3, *p* < 0.05) (Figure 5). 

## 3. Discussion

During the past 50 years, developments in techniques and materials for the repair of tendon injuries have significantly improved the clinical outcomes of patients after surgery. Unfortunately, excellent results are not yet attained universally [15,16]. Tendon healing depends on the ability of the injured tendon to recruit cells from its surface for intrinsic healing. As the tendon is a hypercellular tissue, tendons require a much longer period of time to heal than other connective tissues [17,18]. Understanding the mechanism of tenocyte proliferation and migration is crucial to find ways to accelerate tendon intrinsic healing. Inflammatory factor and growth factor stimuli have been reported to play an important role in the induction of cell proliferation and differentiation, cell alignment and migration, extracellular matrix synthesis, and tissue remodeling in all tendon healing processes [19,20,21]. Tenocyte proliferation and migration are fundamental for tendon development, homeostasis, and regeneration after injury. In our study, we found that after 24 h in tissue culture, T-ABRM significantly increased cell proliferation and migration in both tenocytes and BMDSCs compared to other groups including T-ABDM, GDF-5, and basal media. These results indicated that the cell vesicles between 1 and 3 µm, which mainly contain apoptotic bodies, have more significantly positive effects on tenocytes and BMDSCs compared to the vesicles less than 1 um of the T-ABDM media. However, it is unknown which proteins, cytokines, and/or growth factors are contained in the apoptotic bodies that ranged from 1 to 3 µm. 

Traditionally, apoptosis has been regarded as the silent cell death because it does not trigger an inflammatory response. However, a number of recent studies have shown some evidence of paracrine signals originating from apoptotic cells [22,23]. Using several different model systems, some studies maintained that the appearance of apoptotic cells can represent a signal for the proliferation of stem or progenitor cell populations, in which the compensatory proliferation was vital for the repair and regeneration of injured tissue [8,24,25,26]. In our flow cytometry analysis, we found that in T-ABRM, the majority of vesicles were between 1 and 3 µm in size, which may include apoptotic bodies and large proteins. After high-speed centrifugation, a large portion of the apoptotic body was depleted, leaving the vesicles smaller than 1 µm in the medium. Although ABDM can also enhance cell proliferation and migration, its effects were significantly less than with ABRM. 

BMDSCs have been found to exert therapeutic effects not only by direct differentiation into surrounding wound tissues, but also by the production of autocrine and paracrine factors [27,28]. The application of BMDSCs has been reported to enhance tendon healing [29]. Our results showed that apoptotic bodies could further enhance BMDSC and tenocyte proliferation and migration.

Some cell types, such as thymocytes and neutrophils, have been reported not to produce apoptotic bodies [14]. However, in our study, tenocytes like endothelial cells have the ability to undergo apoptotic body formation. In our study, fluorescence-activated cell analysis showed that tenocytes had gone through apoptosis and produced a large amount of apoptotic bodies after 24 h of serum deprivation, which is consistent with the results of another study using endothelial cells [30]. Immunohistochemistry also showed that tenocyte apoptotic bodies contain intact membrane stained with DiI and part of the nucleus DNA stained with DAPI (Figure 1C).

TGF-β is a secreted protein that controls proliferation, cellular differentiation, and migration [31,32]. TGF-β expression of both tenocytes and BMDSCs increased after ABRM stimulation. Expression of types I and III collagen also increased after ABRM treatment. This may be due to the phenotypic differences between the two different cell lines. 

## 4. Materials and Methods

### 4.1. Isolation of Apoptotic Bodies Derived from Tendon Cells

Tenocytes were isolated and cultured with protocols performed as previously described [33]. Tenocytes in passage 3–5 were incubated for 24 h in basal media without serum to induce apoptosis [30]. Conditioned media from apoptotic tenocytes were centrifuged (800 g for 10 min) to discard dead cells and large debris from the media to obtain the T-ABRM based on the established protocol [30]. This T-ABRM was further centrifuged (16,000 g for 20 min) to deplete the apoptotic bodies to obtain T-ABDM (Figure 6) [30]. Aannexin V/fluorescein isothiocyanate (FITC; BD PharMingen, Hamburg, Germany) and PI (BD PharMingen, Hamburg, Germany) were used to identify the cell’s apoptotic and necrotic responses. The flow cytometry with a single-cell gate was performed and analyzed in a fluorescence-1/fluorescence-2 dot plot to quantify the percentage of annexin V+/PI− cells, representing the apoptotic and necrotic populations.

### 4.2. Characterization of T-ABRM by Flow Cytometry and Immunohistochemistry

T-ABRM was analyzed by flow cytometry choosing forward/side scatter dot in a fluorescence-activated cell sorter (FACScan, BD FACSCanto). For flow cytometry, apoptotic bodies were stained for 30 min with an annexin V-FITC apoptosis detection kit (1:400) as per the previous manufacturer’s instructions (BD Pharmingen, BD Bioscience, USA). In addition, tenocytes were incubated with 1 g/mL PI before flow cytometry for 10 min at room temperature. To measure the size, we used a size marker (1.0 µm, 3.0 µm, 6.0 µm; Bangs Laboratories, Inc., Fishers, IN, USA). For tenocytes, apoptotic bodies were close to the range of 1.0 to 3.0 µm, whereas microvesicles were much smaller than 1 µm [34]. For fluorescence microscopy, after 24 h of serum deprivation, the DiI-stained tenocytes (the tenocytes were pre-stained with 5 ul/mL DiI [1,1′-Dioctadecyl-3,3,3′,3′-Tetramethylindocarbocyanine Perchlorate, Sigma-Aldrich Inc., St. Louis, MO, USA] at 37 degrees Celsius for 20 min and then washed with PBS) were plated on coverslips fixed with 4% paraformaldehyde and stained with 1 g/mL DAPI (4′,6-diamidino-2-phenylindole dihydrochloride; Sigma, St. Louis, MO, USA). Images from the fluorescence microscope were taken using a confocal imaging system (LSM 780 microscope system, Zeiss, Oberkochen, Germany).

Each group with normal tenocytes were starved of serum for 24 h and labelled with Vybrant Cell-Labeling Solutions (Molecular Probes; Life Technologies, Carlsbad, CA, USA) according to the manufacturer’s instructions before tenocytes were seeded on cover slices. Cell-seeded coverslips were cultured in 6-well plates until reaching 50–60% confluence (about 6 h). Then, cells were fixed with fresh 4% paraformaldehyde for five minutes and then washed three times with phosphate-buffered saline solution. To mount cells on a slide, VECTASHIELD Mounting Medium with DAPI (Vector Laboratories, Inc., Newark, CA, USA) was dispersed over the entire section according to the instructions. Tenocytes were observed with a confocal microscope (LSM 780; Zeiss, Oberkochen, Germany).

### 4.3. Measurement of Cell Proliferation and Viability

The isolation and culture of BMDSCs were performed using previously described protocols [35]. To evaluate tenocyte and BMDSC viability and proliferation, the Cell Counting Kit-8 (Dojindo Molecular Technologies, Inc., Rockville, MD, USA) was used for each group. This assay was performed according to the manufacturer’s instructions. BMDSCs and tenocytes were first cultured in a series of known quantities (0.1 × 10^3^–1.0 × 10^6^) in a 2-fold linear dilution ratio in 96-well plates and absorbance cell number standard curves were used to calculate quantity. Then, the conditioned media co-cultured cells were cultured in different culture times (0, 12, 24, 48, and 72 h) and the absorbance optical density value at 450 nm was tested and determined with an enzyme-linked immunosorbent assay reader. The total number of cells in each group was calculated according to the standard curve. 

### 4.4. Measurement of Cell Migration

Tenocytes passaged 2–3 were incubated for two hours in appropriate serum-free media (MEM only) prior to cell migration assay. Then, with 0.25% trypsin-EDTA (Gibco; Thermo Fisher Scientific, Waltham, MA, USA), cells were harvested and washed twice. Cells should be aspirated by pipetting up and down gently; it is important to break down into individual cells as much as possible. Trypsin and inhibitors were removed by spinning down the cells with MEM and 0.5% FBS, and then the cells were resuspended in MEM with 0.5% FBS and counted. The Millipore QCM cell migration assay kit was prepared in a 24-well format with 8-μm pore size inserts according to the user instructions. Then 1 × 10^5^ cells were gently added to the upper compartment of the inserts. The cells were incubated in Transwell plates (Corning, NY, USA) at 37 °C and 5% carbon dioxide for 2.5 h. This allows cells to migrate toward the underside of the insert filter (Figure 7). After 24 h of cell culture in different co-cultured media, the insert was carefully removed. Cells that did not migrate through the membrane remained on the upper side of the filter membrane and were gently removed with a cotton swab. Cells on the lower side of the insert filter were quickly fixed by 5% glutaraldehyde for 10 min and then stained with a cell stain medium for 20 min. The non-migratory cell layer was removed from the interior of the insert. Excess water was drained from the side of the insert using a cotton swab to keep the insert membrane dry. The cells on the lower side of the filter were imaged under a microscope and recorded. The stained insert was transferred to a clean well containing 200 μL of extraction buffer for 15 min at room temperature. The stain was extracted from the underside by gently tilting the insert back and forth several times during incubation. The insert was removed from the well. The dye mixture was then transferred to a 96-well microplate suitable for colorimetric measurement. The optical density at 560 nm was measured. The same experimental procedure was performed for control groups with only MEM with 2% FBS and GDF-5 (100 ng/mL) [36]. Each migration condition was tested three times.

### 4.5. RT and Quantitative Real-Time PCR 

Total RNA was obtained from cultured BMDSCs and tenocytes and extracted using TRIzol reagent (Invitrogen, Waltham, MA, USA) according to the manufacturer’s protocol. The total RNA concentration was determined using a NanoDrop (Thermo Scientific, Waltham, MA, USA). cDNA was synthesized using a Transcriptor First Strand cDNA Synthesis Kit (Roche, Basel, Switzerland) with anchored oligo (dT) primer. The reverse transcriptase was inactivated by heating to 85 °C for 5 min. The quantitative RT-PCR test was performed using a Light Cycler 480 SYBR Green I Master kit (Roche) in a LightCycler 480 instrument (Roche). The following amplification cycles were employed for all genes: 5 min of an initial denaturation at 95 °C, followed by 45 cycles of 95 °C, 60 °C, and 72 °C for 10, 20, and 20 s, respectively, plus an extension at 72 °C for 5 min. Those samples were measured in each group.

### 4.6. Statistical Analysis

The results of the migration, proliferation, and gene expression studies were analyzed by one-way factorial analysis of variance. A Student’s *t* test was used for the statistical analysis. A Tukey-Kramer post hoc test for each pairwise comparison was performed if there was a significant difference. All results were shown as means, with the SD in parentheses. The significance level was set to *p* < 0.05 in all cases. All statistical analyses were performed using JMP software, version 9.0.1 (SAS Institute, Cary, NC, USA).

## 5. Conclusions

In conclusion, our data indicate that tenocytes and BMDSCs cultured in ABRM or AMDB from tenocytes enhanced cell proliferation, migration, and production, in which ABRM showed better results compared to ABDM. Our findings have a significant impact in two respects. First, the findings explore the possible mechanism of intrinsic healing of the tendon in which tenocyte proliferation and migration may be initiated by the apoptosis and necrosis of the tenocytes following tendon injury. Second, our findings may provide basic support for the potential novel therapeutic treatments to enhance tendon healing using major cytokines contained in the T-ABRM or T-ABDM after they are identified with proteomic arrays in our future studies. This could open up a new arena for tendon healing augmentation, especially when BMDSCs are considered as a cell-based treatment.

## Figures and Tables

**Figure 1 ijms-23-11475-f001:**
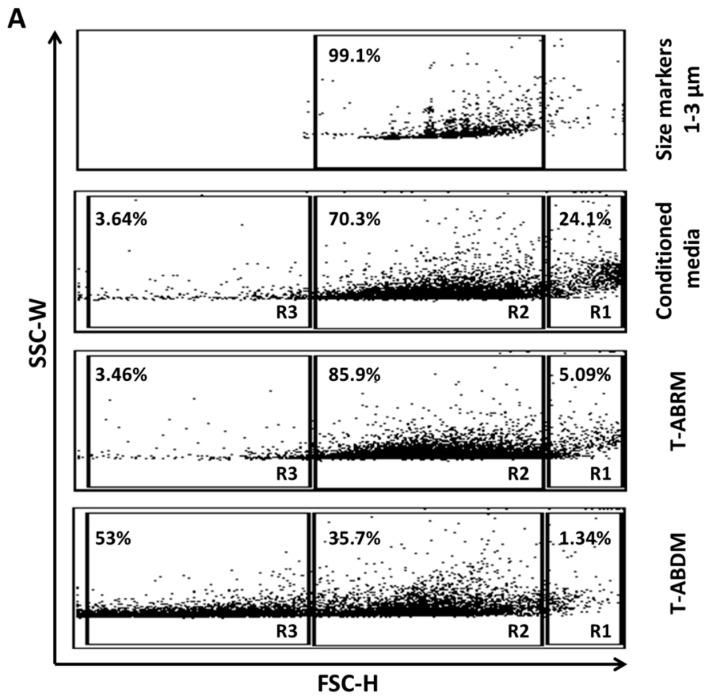
Tenocyte-derived apoptotic particles characterized by flow cytometry. (**A**) FSC/SSC dot plot analysis of particles from apoptotic tenocytes. Size markers were used at 1.0, 3.0, and 6.0 µm. Conditioned media from apoptotic tenocytes contain suspended cells and dead cell debris (gate R1), apoptotic bodies or vesicles between 1 and 3 µm (gate R2), and microvesicles smaller than 1µm (gate R3). T-ABRM, obtained by centrifugation (800 g, 10 min), contains the most apoptotic bodies and vesicles (1–3 µm). T-ABDM, obtained by the centrifugation (16,000 g, 20 min) of T-ABRM, contains mainly small vesicles and rare apoptotic bodies (<1 µm). (**B**) Annexin V/fluorescein isothiocyanate (FL-1) and propidium iodide (FL-2) dot plot analysis of live cells, dead cells, and cell-derived vesicles. Dead cells, T-ABRM, and T-ABDM stained positive with annexin V, but live cells (in red) stained a low binding of annexin V and propidium iodide. The set of quadrant gates was based on the respective unstained control population. The percentage of events or dots in the graph is shown in the upper right corner of the respective region. (**C**) Fluorescence microscopy of DAPI+ and DiI+ tenocytes before and after 24 h of serum-starved treatment demonstrated tenocyte apoptosis in which tenocyte cytoplasm shrunk and dark and unclearly stained nuclei was observed (middle). In addition, many divided nuclei (in green) with cell membrane-covered vesicles (apoptotic bodies in red) were detected (right) (Scale bar was 5 µm) (Representative plots and images from 4 independent experiments).

**Figure 2 ijms-23-11475-f002:**
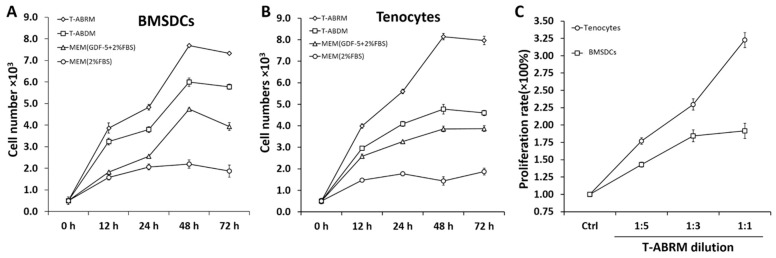
Effect of T-ABRM, T-ABDM, and GDF-5 on the proliferation of tenocytes and BMDSCs. Cell proliferation assay with Cell Counting Kit-8 of tenocytes (**A**) and BMDSCs (**B**). The cell population-absorbance standard curve of each cell type was calculated and analyzed as previously. According to the standard curves, the absorbance of each sample was converted into the cell population. The graph shows the population-time of the cell growth curve with each cell type. (**C**) Line graph depicts the tenocyte and BMDSC proliferation rate after treatment with different dilutions of T-ABRM over 48 h. The protein content of T-ABRM (1:1) was about 0.3 mg/mL. MEM was used to obtain and dilute the T-ABRM. MEM with 2% FBS should only be used as a control. Each sample was examined in quadruplicate. BMDSCs indicate bone marrow stromal cells; Ctrl, control; FBS, fetal bovine serum; GDF-5, growth and differentiation factor 5; MEM, minimum essential media; T-ABDM, tenocyte apoptotic body-depleted media; T-ABRM, tenocyte apoptotic body-rich media.

**Figure 3 ijms-23-11475-f003:**
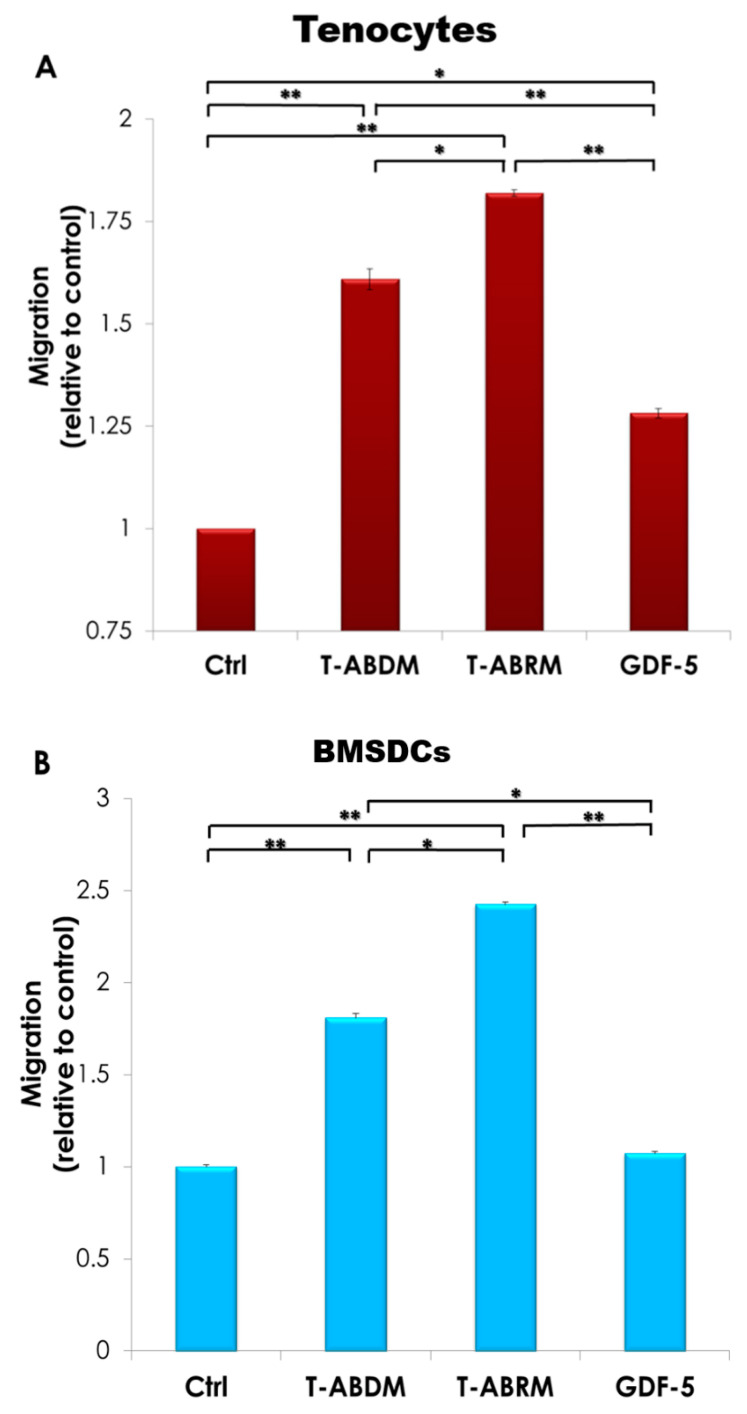
Tenocyte migration test under the T-ABRM stimulated condition. (**A**) Tenocytes were stimulated under different culture conditions, including control (MEM + 2% FBS), T-ABRM, T-ABDM, and GDF-5 (MEM + 2% FBS + 100 ng/mL GDF-5) at 24 h. (**B**) BMDSCs were stimulated under different culture conditions, including control (MEM + 2% FBS), T-ABRM, T-ABDM, and GDF-5 (MEM + 2% FBS + 100 ng/mL GDF-5) at 24 h. Colorimetric measurements were taken according to instructions with an enzyme-linked immunosorbent assay reader. Tenocyte (**C**–**F**) and BMDSC (**G**–**J**) migration assays with the following treatments: control media (**C**,**G**), T-ABDM (**E**,**I**), T-ABRM (**F**,**J**), and GDF-5 media (**D**,**H**) after 24 h. Cell concentration was 10^5^ cells per well. Migrated cells on the bottom side of the membrane were stained and imaged according to assay instructions. * *p* < 0.05, ** *p* < 0.01; *n* = 4, BMDSCs indicate bone marrow stromal cells; Ctrl, control; FBS, fetal bovine serum; GDF-5, growth and differentiation factor 5; MEM, minimum essential media; T-ABDM, tenocyte apoptotic body-depleted media; T-ABRM, tenocyte apoptotic body-rich media.

**Figure 4 ijms-23-11475-f004:**
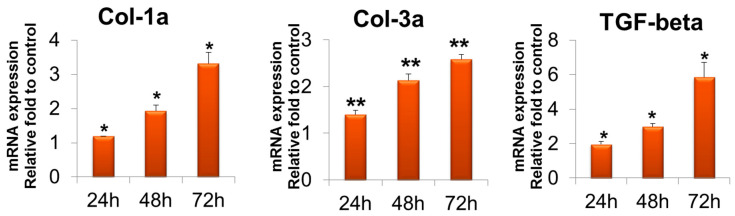
Tenocyte gene expression of Col-1a, Col-3a, and TGF-β in T-ABRM treatment. * *p* < 0.05, ** *p* < 0.01, compared with the control. Col-3a indicates collagen 1a; Col-3a, collagen 3a; T-ABRM, tenocyte apoptotic body-rich media; TGF-β, transforming growth factor β.

**Figure 5 ijms-23-11475-f005:**
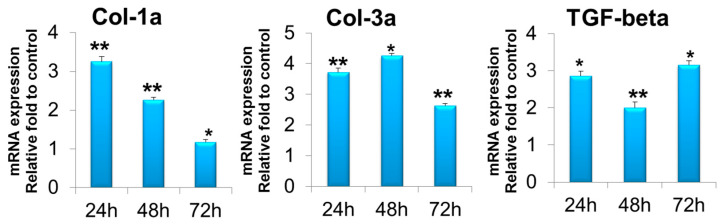
BMDSC gene expression of Col-1a, Col-3a, and TGF-β in T-ABRM treatment. * *p* < 0.05, ** *p* < 0.01, *n* = 3, compared with the control. BMDSCs indicate bone marrow stromal cells; Col-1a, collagen 1a; Col-3a, collagen 3a; T-ABRM, tenocyte apoptotic body-rich media; TGF-β, transforming growth factor β.

**Figure 6 ijms-23-11475-f006:**
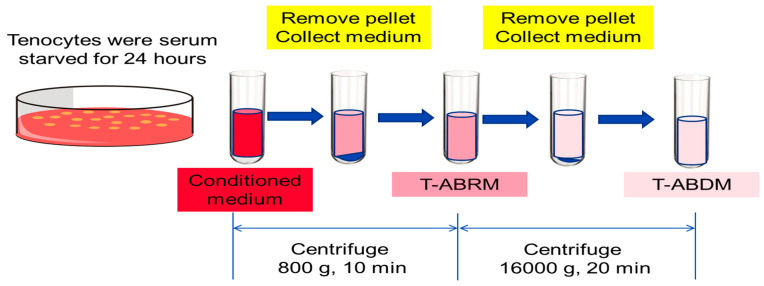
Diagram of the generation of tenocyte apoptotic body-rich media (T-ABRM) and tenocyte apoptotic body-depleted media (T-ABDM) from conditioned media.

**Figure 7 ijms-23-11475-f007:**
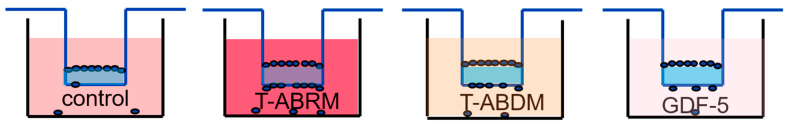
Schematic diagram of the cell migration assay. GDF-5 indicates growth and differentiation factor 5; T-ABDM, tenocyte apoptotic body-depleted media; T-ABRM, tenocyte apoptotic body-rich media.

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
