# Peer review of "Apoptotic Body-Rich Media from Tenocytes Enhance Proliferation and Migration of Tenocytes and Bone Marrow Stromal Cells"

_ijms, 2022, doi:10.3390/ijms231911475_

Round 1
Reviewer 1 Report
In the title (Address) what is USC stand for?
How ABRM maintain pluripotency in tenocytes and Bone marrow stromal cells (BMSC) is not understood. What is a purpose of measuring these markers which are significantly low in expression. Moreover, it is wrong to say that ABRM increases BMSC stemness. This statement contradicts the purpose of this experiment.
Rather authors should have checked Scleraxis or Tenomodulin markers in order to understand whether apoptotic body rich conditioned media from tenocytes increase proliferation, migration and differentiation during tendon injury.
I would also suggest to check Alkaline phosphate and BSP markers.
What's a purpose of using GDF-5 group? It is far from understanding. Perhaps purpose of using GDF-5 may be to provide growth factor in MEM to check proliferation and migration in comparison to ABRM and ADCM. It is therefore more appropriate to check growth factor profile present in the both conditioned media as they significantly proliferate and migrate tenocytes and BMSC.
In the text authors mentioned GDF-5 but in Figure-7 it is mentioned as GFP-5?
In the conclusion authors mentioned that tenocytes and BMSC cultured in ABRM enhanced proliferation, migration and production. What is this production?
Overall, there is a flaw in the experimental design but that can be improved upon few additional experiments.
Author Response
- How ABRM maintain pluripotency in tenocytes and Bone marrow stromal cells (BMSC) is not understood. What is a purpose of measuring these markers which are significantly low in expression. Moreover, it is wrong to say that ABRM increases BMSC stemness. This statement contradicts the purpose of this experiment.
Response: Thank you for the comment. We agree with the reviewer that ABRM maintain pluripotency in tenocytes and BMSC is unclear. The purpose of measuring these markers intended to understand if the ABRM could help tenocytes to de-differentiate into stem-cell-like cells for the tendon regeneration. However, this could cause the confusion of the purpose of the study. Therefore, we have eliminated these data from the manuscript to avoid confusion. We have also deleted that sentence in the text as suggested by the reviewer.
- Rather authors should have checked Scleraxis or Tenomodulin markers in order to understand whether apoptotic body rich conditioned media from tenocytes increase proliferation, migration and differentiation during tendon injury. I would also suggest to check Alkaline phosphate and BSP markers.
Response: Thank the reviewer’s suggestion. We did not check the osteogenic markers in this manuscript since we focused on the tenogenesis. However, this should be done to see if the ABRM could also promote other differentiation potential beside the tenogenesis. Therefore, we have listed this as one of the study limitations.
- What's a purpose of using GDF-5 group? It is far from understanding. Perhaps purpose of using GDF-5 may be to provide growth factor in MEM to check proliferation and migration in comparison to ABRM and ADCM. It is therefore more appropriate to check growth factor profile present in the both conditioned media as they significantly proliferate and migrate tenocytes and BMSC.
Response: We thank the reviewer for the careful review and comments. The purpose of using GDF-5 was to serve as a positive control group, since it is known that GDF-5 can enhance the tendon healing by promote cell proliferation, migration, and differentiation. as reference reported. (Hayashi M et al. “The effects of growth and differentiation factor 5 on bone marrow stromal cell transplants in an in vitro tendon healing model, J Hand Surgy (Eur) 36(4):271-9, 2011). We have clarified in the text and site the reference.
- In the text authors mentioned GDF-5 but in Figure-7 it is mentioned as GFP-5?
Response: We are sorry for the mistake, we have corrected this error in the revision. Thank you for the careful review!
- In the conclusion authors mentioned that tenocytes and BMSC cultured in ABRM enhanced proliferation, migration and production. What is this production?
Response: The production here is indicating cellular production that could enhance tendon healing. The Col-1 and 3 are the major cellular production for tenocytes for tendon healing. We have clarified the confusion in the revision.
- Overall, there is a flaw in the experimental design but that can be improved upon few additional experiments.
Response: We do agree with the reviewer for this comment. We have listed the reviewer’s important points as the study limitations that could be addressed in our future studies. Thank you!

Reviewer 2 Report
The paper by Dong et al. presents a study of proliferation and migration of tenocytes and MSCs using an apoptotic body–rich media from Tenocytes. The experiments appear to have been done with skill and care. However, the problem in this study is rationale for the experimental design. Some of their conclusions are also not supported by their experimental data. Moreover, the results are insufficient to reveal any mechanism or draw any significant conclusion, which dampens the enthusiasm. In general, the study is quite preliminary and the data are rather limited.
1. In the Introduction section, the authors should make an effort to address why they investigate the effect of apoptotic body derived from tenocytes on proliferation and migration of tenocytes and MSCs.
Moreover, there is a possibility that phenotype of MSCs were changed after treatment with apoptotic body derived from tenocytes. Authors need to show characterization of cells both before and after treatment.
2. The authors showed that T-ABRM and T-ABDM both treatments significantly stimulated the proliferation, migration for both tenocytes and MSCs compared to the control groups, in this case, how did they draw the conclusion that T-ABRM enhances the proliferation and migration of tenocytes and MSCs?
3. The authors showed that T-ABRM treatment increases proliferation and migration in tenocytes and MSCs, they concluded that T-ABRM significantly stimulated the cell migration. The conclusion may not be correct, because the higher migration might be the result of proliferation increase by T-ABRM treatment.
4. In Fig.4/5, gene expression of Col-1a, Col-3a, TGF-β, Nanog, and Sox-2 mRNA in T-ABRM treatment was detected. What is the purpose of this experiment? What is the relation between these gene expression and the proliferation and migration of tenocyte and MSCs?
5. The authors hypothesized that secretions from injured tenocytes might have biological effects on cell proliferation and migration to enhance tendon healing. In theory, the information of T-ABRM effect on in vivo tendon healing should be included in this study. Do the authors have this piece of information?
Author Response
The paper by Dong et al. presents a study of proliferation and migration of tenocytes and MSCs using an apoptotic body–rich media from Tenocytes. The experiments appear to have been done with skill and care. However, the problem in this study is rationale for the experimental design. Some of their conclusions are also not supported by their experimental data. Moreover, the results are insufficient to reveal any mechanism or draw any significant conclusion, which dampens the enthusiasm. In general, the study is quite preliminary and the data are rather limited.
Response: We thank the reviewer for the careful review and constructive comments. We agree with the reviewer that this study is a preliminary report with limited data. We did not analyze what ABRM or ABDM contains which could help us to understand the mechanism. We also apologize for some unclear and confusion points in the first submission. We have revised the manuscript to clarify some confusions and study limitations that the reviewer raised. We hope this revision could satisfy the reviewer.
- In the Introduction section, the authors should make an effort to address why they investigate the effect of apoptotic body derived from tenocytes on proliferation and migration of tenocytes and MSCs. Moreover, there is a possibility that phenotype of MSCs were changed after treatment with apoptotic body derived from tenocytes. Authors need to show characterization of cells both before and after treatment.
Response: We thank the reviewer for the comment. We have revised the Introduction section to clarify the rationale of the study with the following paragraph “One evidence for tendon intrinsic healing is that the tenocytes or progenitor cells proliferate and migrate into the injury site to promote tendon healing by producing Col-1 and Col-3. However, the mechanism of this cellular activities is unclear. It is known that tenocytes at the injury site undergo apoptosis after tendon injury. Apoptotic body formation occurs when cells undergo apoptosis. This packaging of apoptotic cells is primarily thought to mediate immunologic and phagocytotic clearance. However, whether the secretions from apoptotic cells would stimulate the cell migration and proliferation of intrinsic tenocytes or progenitor cells from either blood or bone marrow are unknown.” (page 2, line 48)
- The authors showed that T-ABRM and T-ABDM both treatments significantly stimulated the proliferation, migration for both tenocytes and MSCs compared to the control groups, in this case, how did they draw the conclusion that T-ABRM enhances the proliferation and migration of tenocytes and MSCs?
Response: We apologize about this confusion. Yes, both ABRM and ABDM significantly stimulate the cell proliferation and migration. However, compared ABRM and ABDM, we saw the ABRM increased the cell proliferation and migration significantly than the ABDM. We have revised the conclusion to clarify this point as the following: “In conclusion, our data indicate that tenocytes and BMDSCs cultured in ABRM or ABDM from tenocytes enhanced cell proliferation, migration, and production, in which ABRM showed better results compared to ABDM. Our findings have a significant impact in two respects. First, the findings explore the possible mechanism of tendon intrinsic healing in which tenocyte proliferation and migration may be initiated by the apoptosis and necrosis of the tenocytes following tendon injury. Second, our findings may provide basic support for the potential novel therapeutic treatments to enhance tendon healing using major cytokines contained in the T-ABRM or T-ABDM after they are identified with proteomic array in our future studies. This could open a new arena for tendon healing augmentation, especially when BMDSCs are considered as a cell-based treatment.” (page 11, line 323)
- The authors showed that T-ABRM treatment increases proliferation and migration in tenocytes and MSCs, they concluded that T-ABRM significantly stimulated the cell migration. The conclusion may not be correct, because the higher migration might be the result of proliferation increase by T-ABRM treatment.
Response: We agree with the reviewer that increased cell migration may be related to the increased cell proliferation. However, we did not use a typical scratch technique for measuring cell migration, which may related to the cell population. We used permeable Transwell plates (Corning, NY) (Fig 7 showed below) to detect cell migration, which is more related to the cell mobility. Therefore, we think our conclusion is correct.
- In Fig.4/5, gene expression of Col-1a, Col-3a, TGF-β, Nanog, and Sox-2 mRNA in T-ABRM treatment was detected. What is the purpose of this experiment? What is the relation between these gene expression and the proliferation and migration of tenocyte and MSCs?
Response: Thank you for the careful review and comments. The purpose of measuring those genes was to determine if tenogenesis related markers (Col-1, Col-3, and TGF-b). The purpose of measuring Nanog and Sox-2 is to understand if the ABRM could help tenocytes to de-differentiate into stem-cell-like cells for the tendon regeneration. However, this could cause the confusion of the purpose of the study as the first reviewer pointed out. Therefore, we have eliminated these data from the manuscript to avoid confusion.
- The authors hypothesized that secretions from injured tenocytes might have biological effects on cell proliferation and migration to enhance tendon healing. In theory, the information of T-ABRM effect on in vivotendon healing should be included in this study. Do the authors have this piece of information?
Response: We agree with the reviewer that in vivo study to validate if T-ABRM indeed could enhance tendon healing would be greatly strength the study. We will pursue the investigation alone this research in the near future. Thanks!

Reviewer 3 Report
Very interesting results. News from the apoptotic world, probably exosomes eventhough size of the particles were higher than exosome sizes. Clear results, just one remark, I would always report the number of experiments performed for every data presented.
Author Response
Very interesting results. News from the apoptotic world, probably exosomes even though size of the particles were higher than exosome sizes. Clear results, just one remark, I would always report the number of experiments performed for every data presented.
Response: We thank the reviewer for the careful review and encouraging comments. We have added the sample number in the manuscript.

Round 2
Reviewer 1 Report
Manuscript is duly modified as per suggestions.
Reviewer 2 Report
The authors have made efforts to revise the manuscript and have responded to this Reviewer's comments. Their revision is appropriate and the response to my comments is satisfactory.